# Differential modularity of the mammalian *Engrailed 1* enhancer network directs sweat gland development

**Daniel Aldea**[1], **Blerina Kokalari**[1], **Yuji Atsuta**[2¤], **Heather L. Dingwall** [1], **Ying Zheng**[3], **Arben Nace**[3], **George Cotsarelis**[3], **Yana G. Kamberov** [1] *

**1** Department of Genetics, Perelman School of Medicine, Philadelphia, Pennsylvania, United States of America, **2** Genetics Department, Harvard Medical School, Boston, Massachusetts, United States of America, **3** Department of Dermatology, Perelman School of Medicine, Philadelphia, Pennsylvania, United States of America

¤ Current address: Department of Biology, Kyushu University, Fukuoka, Japan
* yana2@pennmedicine.upenn.edu

**Data Availability Statement:** All data are available in the manuscript or the supplementary materials.

**Funding:** Funding for this study was supported by award P30-DK19525 (https://www.nih.gov/) to the

## Abstract

Enhancers are context-specific regulators of expression that drive biological complexity and variation through the redeployment of conserved genes. An example of this is the enhancer-mediated control of *Engrailed 1* (*EN1*), a pleiotropic gene whose expression is required for the formation of mammalian eccrine sweat glands. We previously identified the *En1* candidate enhancer (ECE) 18 *cis*-regulatory element that has been highly and repeatedly derived on the human lineage to potentiate ectodermal *EN1* and induce our species' uniquely high eccrine gland density. Intriguingly, ECE18 quantitative activity is negligible outside of primates and ECE18 is not required for *En1* regulation and eccrine gland formation in mice, raising the possibility that distinct enhancers have evolved to modulate the same trait. Here we report the identification of the ECE20 enhancer and show it has conserved functionality in mouse and human developing skin ectoderm. Unlike ECE18, knock-out of ECE20 in mice reduces ectodermal *En1* and eccrine gland number. Notably, we find ECE20, but not ECE18, is also required for *En1* expression in the embryonic mouse brain, demonstrating that ECE20 is a pleiotropic *En1* enhancer. Finally, that ECE18 deletion does not potentiate the eccrine phenotype of ECE20 knock-out mice supports the secondary incorporation of ECE18 into the regulation of this trait in primates. Our findings reveal that the mammalian *En1* regulatory machinery diversified to incorporate both shared and lineage-restricted enhancers to regulate the same phenotype, and also have implications for understanding the forces that shape the robustness and evolvability of developmental traits.

## Author summary

Enhancers are regulatory elements in the genome that modulate the expression of protein-coding genes by directing how much, where, or when a given gene is expressed. Accordingly, enhancers are major determinants of mammalian traits and thought to be

University of Pennsylvania Diabetes Research Center for the use of the Transgenic and Chimeric Mouse Facility Core to support work in this study, by award P30-AR069589 (https://www.nih.gov/) to the Penn Skin Biology and Diseases Resource-based Center (SBDRC) Cores A and B to support the work in this study, by award 5T32AR007465 (https://www.nih.gov/), which provided salary support to H. L. D., by awards BCS-1847598 (https://www.nsf.gov/) and R01AR077690 (https://www.nih.gov/) Y.G.K., which both provided salary support to Y.G.K. and funds for the work in this study. The funders had no role in study design, data collection and analysis, decision to publish, or preparation of the manuscript.

**Competing interests:** The authors have declared that no competing interests exist.

the predominant drivers of evolutionary change. Here we interrogated the identity and compared the functionality of the enhancers that control the specification of a single, highly variable trait, the density of sweat glands in mammalian skin, by regulating expression of the mammalian *Engrailed 1* (*En1*) gene during development. We find that mammals have evolved two distinct types of enhancers to regulate this single trait: a shared enhancer active in multiple mammalian species that not only controls *En1* expression in the skin but also in the brain, and also an enhancer whose activity is restricted to the skin of primates and rapidly evolved on this lineage to affect sweat phenotypes. Our findings implicate differences in the intrinsic properties of enhancers, namely the extent to which their activity is restricted to a specific context, in shaping not only the complexity of the regulatory landscape of a developmental gene but also the means by which that landscape evolves to generate trait variation.

## Introduction

Eccrine sweat glands are specialized exocrine appendages of mammalian skin and are a major evolutionary innovation of this phylogenic class [1,2]. The main function of eccrine glands is to secrete hypotonic water onto the skin surface in response to specific neurological stimuli [2,3]. In the ancestral and most common state among mammals, eccrine glands develop only in the palmar/plantar (volar) skin of the ventral autopod (the distal-most segment of the limb) [1–3]. Here, eccrine glands are stimulated by cues from the limbic system and their secretions regulate frictional contact and grip [2–7]. In this context, eccrine glands are described as effectors of the fight or flight response, however, recent evidence also suggests that variation in volar eccrine glands is correlated with differential climbing ability in rodents [3,4,8,9]. Eccrine gland distribution has been expanded to nearly the entire body surface in humans and other catarrhine primates (the monkeys of Africa and Asia, and the anthropoid apes) [1,2,10]. Though structurally indistinguishable from their volar counterparts, generalized eccrine glands are activated by signals from the thermosensory regions of the hypothalamus and function in temperature regulation [2,3,10–15]. The physiological importance of this regional expansion is paramount in humans, who have by far the greatest eccrine gland density of any primate [2,10,16], and who are reliant on the vaporization of eccrine sweat as the main means for cooling the body [2,3,5]. Given this extensive diversification, the study of how eccrine gland phenotypes evolved along mammalian lineages provides an exceptional inroad to understand the broader principles through which evolution generates natural variation and functional novelty.

The ability of any region of the skin to build eccrine glands relies on coordinated and reciprocal signaling between the deepest (basal) layer of the skin ectoderm from which eccrine glands arise and the underlying dermal mesenchyme [7,17–21]. In both humans and mice, expression of the transcription factor *Engrailed 1* (*En1*) in the basal ectoderm is a universal hallmark of all skin in which eccrine glands develop [17,22,23]. Moreover, the focal upregulation of *En1* expression is the earliest, specific signature of eccrine gland placodes, thickenings of the ectoderm from which eccrine glands derive [17,22]. In mice, disruption of *En1* expression in the basal ectoderm at the time of placode formation leads to dose-dependent decreases in the number of volar eccrine glands formed [22–24]. Moreover, natural differences in volar eccrine gland number between inbred mouse strains are primarily the product of strain-specific variation in the quantitative levels of ectodermal *En1* [22]. These findings illustrate the critical role of precise spatial, temporal, and quantitative regulation of the *En1* locus in the generation of eccrine phenotypes.

For *En1*, as for the majority of protein coding loci, gene expression is coordinated by regulatory elements, or enhancers, that define where, when, and how much the *En1* promoter is activated. Notably, *En1* is essential for the development of multiple traits independent of its role in eccrine gland formation. These include patterning of the vertebrate midbrain/hindbrain, the formation of the cerebellum, and the dorso-ventral patterning of the embryonic limb bud [22,23,25,26]. Accordingly, control of *En1* expression is likely to involve multiple enhancers that have, at least in part, non-overlapping, context-specific roles. Consistent with this, a recent study implicated a lncRNA-containing region distal to the *En1* promoter in mice and humans that is required for *En1* expression in the ectoderm during dorso-ventral limb patterning, but which is dispensable for *En1* expression in the brain at the same stage [27]. Regulation of the *En1* locus in the skin ectoderm is also subject to differential temporal control as evidenced by our finding that *En1* upregulation in mouse strains with high eccrine gland density coincides with the period of placode formation in the volar ectoderm [22]. However, during earlier developmental stages, including during dorso-ventral patterning, the volar skin ectoderm of mouse strains with high and with low eccrine gland densities shows equivalent levels of *En1* [22].

The importance of context-specific *EN1* regulation to eccrine phenotypes is evident in our own species. Humans have the most dramatically derived eccrine phenotype of all mammals, having evolved an eccrine gland density that is on average ten times that of other catarrhines [2,10]. We recently uncovered that this adaptive elaboration is underlain by the evolution of an ectodermal *EN1* enhancer, ECE18 (hg38 Chr2:118309555–118310531) [24]. The human homolog of ECE18 (hECE18) overlaps the human accelerated region (HAR) 2xHAR20, and has undergone rapid evolution on the human lineage [24,28,29]. Following the split from chimpanzees, successive mutation of the human ECE18 homolog (hECE18) has led to the accumulation of ten derived base substitutions that interact epistatically to render hECE18 the most quantitatively potent of all primate enhancer homologs [24]. Analysis of the endogenous capabilities of the hECE18 enhancer in human basal skin cells (keratinocytes) revealed that this element is required for *EN1* expression in this context [24]. The ability of hECE18 to promote the formation of more eccrine glands by upregulating ectodermal *En1* in a hECE18 mouse knock-in provided functional evidence that this element modifies *EN1* expression to induce our species' derived eccrine gland phenotype [24].

Intriguingly, the contribution of *ECE18* to ectodermal *EN1* regulation appears to be restricted to primates. Quantitative comparisons in mouse and human keratinocytes revealed that *ECE18* homologs of species outside of the primate order have little to no enhancer activity [24]. Moreover, knock-out of the endogenous *ECE18* enhancer in mice produces no effect on ectodermal *En1* expression or on eccrine phenotypes in the volar skin of these animals [24]. Whether such species have evolved entirely different enhancers to modulate *En1* in eccrine gland formation or there is differential redundancy in the activity of *En1* enhancers in lineages outside of primates, is unclear. We therefore set out to determine if other enhancers have evolved to regulate ectodermal *En1* expression in the context of eccrine gland development and to define their relationship to *ECE18* in this respect.

## Results

### ECE20 is a developmental enhancer active in skin basal ectoderm

Using skin-specific, transgenic reporter mice, we have previously identified ECE20 (mm10 Chr1: 121176848–121178300) as a candidate *En1* enhancer (ECE) in a screen for conserved, non-coding elements with enhancer activity in the basal ectoderm of eccrine forming skin (Figs 1A and S1A) [24]. ECE20 is located within the mapped human and mouse *En1*

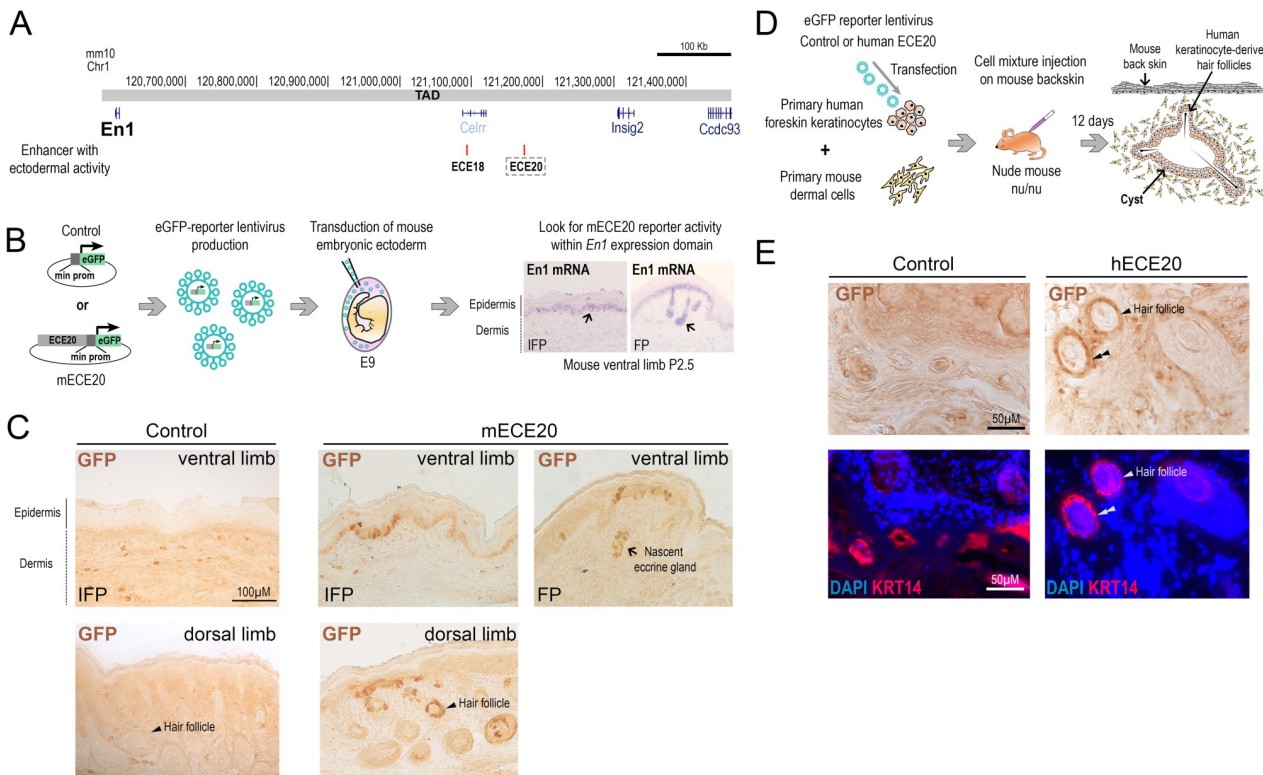

**Fig 1. ECE20 is a developmental enhancer active in basal ectoderm. (A)** Genomic locations of *En1* candidate enhancers (ECEs) ECE18 and ECE20 (boxed) in the mouse genome (mm10). *En1* Topological associating domain (TAD). **(B)** Strategy used to generate skin-specific transgenic mice to assess spatial and temporal activity of mouse ECE20 homolog (mECE20) *in vivo*. *In situ* hybridization for *En1* mRNA (purple) in mouse volar limb skin at P2.5. Representative images of volar skin from the interfootpad (IFP) medial volar skin and footpad (FP) volar skin are shown. Nascent eccrine gland (black arrow). **(C)** Representative images of control and mECE20 transgenic P2.5 volar and dorsal limb autopods stained with anti-GFP antibody. eGFP is visualized using HRP-DAB-coupled immunohistochemistry so signal is brown. Nascent volar eccrine gland (black arrow). Dorsal hair follicle (head arrow). **(D)** Strategy to reconstitute transgenic human-derived developing ectoderm in hybrid human-mouse skin patches to test spatial and temporal activity of human ECE20 (hECE20) *in vivo*. **(E)** Representative images of a section from human developing skin infected with hECE20 or control virus and stained with antibody against GFP (brown) and also stained with the basal keratinocyte specific marker Keratin 14 (KRT14, Red). eGFP (black double head arrow) is visualized using HRP-DAB coupled immunohistochemistry so positive clones are brown. Primary human basal keratinocyte cells double positive for GFP and KRT14 (white double head arrow). DAPI nuclear stain (blue). See also S1 Fig.

topologically associated domains (TADs) as defined by High-C, approximately 500kb downstream of the *En1* locus, and 79kb downstream of the ECE18 enhancer (Figs 1A, S1A and S1B) [24,30]. In contrast to ECE18, ECE20 does not show evidence of accelerated evolution in any mammalian clade [24,28,29].

Consistent with our published findings, the mouse homolog of ECE20 (mECE20) reproducibly induces eGFP reporter expression in the basal keratinocytes of the post-natal day (P) 2.5 mouse ventral autopod, recapitulating the pattern of endogenous *En1* expression that is required at this stage for eccrine gland development (Fig 1B and 1C). In addition, mECE20 but not control transgenic mice have numerous GFP positive clones in the dorsal limb within the skin basal layer and also in the basal cells of the hair follicles that are present in this region (Fig 1C). This finding is significant in light of the expansion of *EN1* expression to the developing basal ectoderm of the non-volar skin in species with generalized eccrine glands such as humans.

Assays to directly evaluate the enhancer activity of catarrhine ECE20 in a relevant context are not available since there are no *in vitro* models that recapitulate eccrine gland development. Moreover *EN1* expression in the generalized basal skin ectoderm of such species begins late in

gestation (~20 weeks gestation in humans), prohibiting direct interrogation of the ECE20 locus *in situ* [17]. Thus, to evaluate the potential of ECE20 as a developmental *EN1* enhancer in the basal ectoderm of generalized skin, we turned to a heterotypic skin reconstitution assay in which primary human keratinocytes are directed to develop stratified epidermis and appendages by inductive developmental cues from mouse dermis (Fig 1D) [31]. To this end, primary human keratinocytes stably infected with lentivirus expressing an eGFP reporter cassette downstream of the human ECE20 (hECE20) enhancer (chr2: 118224252–118225644 [hg38])and minimal promoter, or with control eGFP virus, are mixed with inductive dermal cells from P0 mouse backskin and injected subcutaneously into the backs of nude mice [31]. In the resulting patches of humanized skin, the human-derived basal ectoderm becomes established, stratifies to form the superficial skin layers, and also gives rise to human keratinocyte-derived hair follicles, in accordance with the inductive potential of the underlying mouse dermis (Fig 1E) [31]. After 12 days, multiple eGFP positive clones are evident specifically in the basal, Keratin 14 (KRT14) positive, human keratinocytes of hECE20 infected patches, but not those infected with control virus (Fig 1E). Consistent with our observations in mECE20 transgenic mice, eGFP positive clones are also present in the human basal cells of the developing hair follicles (Fig 1E). This spatial pattern of activity is consistent with the pattern of *EN1* expression in human developing skin, in which *EN1* is expressed throughout the basal ectoderm including in the basal cells of fetal hair follicles. Notably, ECE20 homologs showed very weak or undetectable enhancer activity in cultured keratinocytes, suggesting that ECE20 enhancer activity is context-specific (S1C Fig). Our findings demonstrate that ECE20 homologs can function as enhancers in the *Engrailed 1*-expressing basal ectoderm during development, raising the possibility that ECE20 may be a conserved component of the endogenous machinery that modulates eccrine phenotypes in mammals.

## ECE20 potentiates ectodermal *En1* expression to induce the formation of eccrine glands

To determine whether mECE20 is an ectodermal *En1* enhancer required for the development of eccrine glands we deleted the mECE20-containing genomic interval from the C57BL/6J mouse genome to create a mECE20 knock-out (mECE20KO) and analyzed *En1* expression and eccrine phenotypes (Figs 2A and S2A). Homozygous mECE20KO mice are born at Mendelian ratios, are viable and fertile. Quantitative RT-PCR (qRT-PCR) analysis of *En1* expression in P2.5 mouse volar skin reveals an average decrease of 20% in *En1* mRNA in mECE20KO homozygotes as compared to wildtype (WT) sibling controls (Figs 2B and S2B). At this stage, the mouse volar skin is populated by eccrine gland anlagen at multiple stages of development, ranging from placodes in the central, inter-footpad (IFP) space to invaginating, nascent glands in the footpads (FPs) (Fig 1B) [22,32]. Comparison of the ratios of allelic *En1* transcripts in the P2.5 volar skin of C57BL/6JWT: FVB/N as compared to C57BL/6JmECE20KO: FVB/N F1 hybrid mice shows that transcription is specifically downregulated at the *En1* locus that has the ECE20KO allele *in cis.* (Figs 2C and S2C). Given this activity, we note that *in silico* motif analyses of human and mouse ECE20 homologs predict conserved biding sites for a number of transcription factors with enriched expression in the skin (S2D Fig). These data show that mECE20 is a bona fide *En1* enhancer that upregulates ectodermal expression of this gene in the skin during eccrine gland development.

The importance of mECE20-dependent *En1* upregulation is made evident in the volar skin of adult ECE20KO mice, who have on average 30% fewer eccrine glands in the IFP as compared to WT sibling controls (Figs 2D, 2E and S2E). Our previous studies implicating *En1* in generating strain-specific variation in mouse volar eccrine gland density revealed that the IFP region

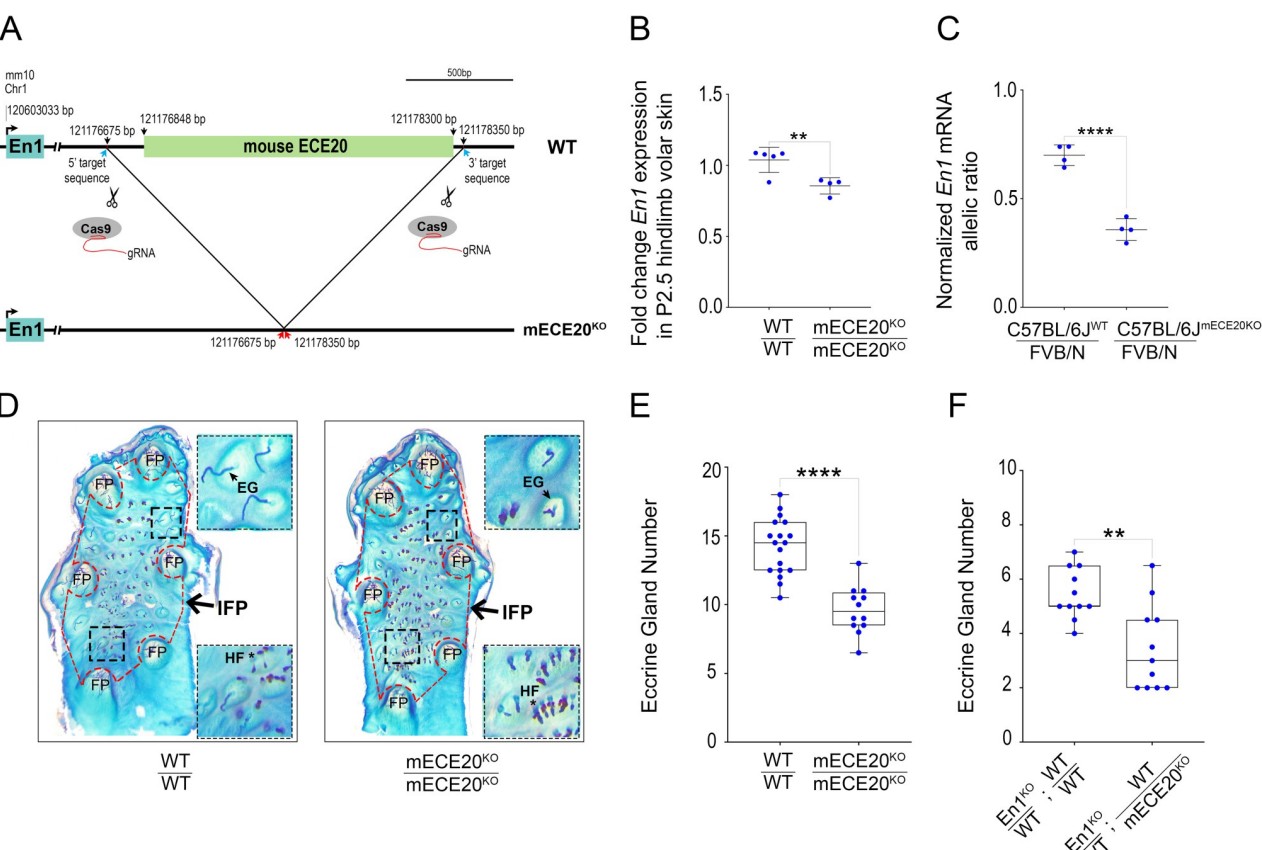

**Fig 2. ECE20 potentiates *En1* expression to induce the formation of eccrine glands. (A)** Strategy to generate mECE20 knock-out (mECE20[KO]) mouse. **(B)** Fold change in *En1* mRNA by qRT-PCR in P2.5 volar hindlimb skin of wildtype (WT / WT), and mECE20[KO] homozygote (mECE20[KO]/ mECE20[KO]) mice relative to mean wildtype value. **(C)** Normalized *En1* mRNA allelic ratio in volar hindlimb skin at P2.5 of wildtype (C57BL/6J[WT] / FVB/N) and mECE20[KO] (C57BL/6J(mECE20[KO]) / FVB/N) hybrid mice. Ratios reported are normalized to the genomic DNA allelic ratio. **(D)** Representative images of whole mount preparations of the volar hindlimb skin from WT / WT and mECE20[KO]/ mECE20[KO] adult mice. Epidermal preparations were stained to facilitate quantification of appendages in the interfootpad area (IFP, outlined in red dashed border). Insets show magnified views of regions within the IFP (black dashed border). Footpad (FP), eccrine gland (EG) and hair follicles (HF). **(E)** Quantification of IFP eccrine glands in the hindlimb volar skin of WT / WT and mECE20[KO]/ mECE20[KO] mice. **(F)** Quantification of IFP eccrine glands in En1[KO] / WT; WT / WT and En1[KO] / WT; WT / mECE20[KO] mice. In **(B, C)** each dot represents the mean value for an individual biological sample. In **(E, F)** each point represents the average number of eccrine glands in the IFP across both hindlimbs of a single mouse. In **(B, C)** mean (line) with standard deviation are plotted. In **(E, F)** the median (line) and maximum and minimum are reported for each genotype. In **(B, C, E, F)** significance assessed by a two-tailed T-test. ****$P<0.0001$, ** $P<0.01$. (KO) knock-out. In **(B)** *Rlp13a* was used as a housekeeping transcript for normalization. See also S2 Fig.

and the eccrine glands that form therein are exceptionally sensitive to *En1* levels [22]. Consistent with a mechanism in which ECE20 effects are mediated by the effect of this enhancer on *En1*, deletion of mECE20 potentiates the reduction in IFP eccrine glands in mice carrying one copy of the *En1* knock-out allele (En1[KO]) (Figs 2F and S2F). These data implicate ECE20 as a conserved, developmental *En1* enhancer necessary for the formation of eccrine sweat glands in the skin.

## ECE20 has pleiotropic effects on *En1* expression

mECE20 directly upregulates ectodermal *En1* during eccrine gland development. Publicly available data from chromosome conformation capture (Capture C) analyses of the mouse embryonic limb bud and brain reveal that the genomic region containing mECE20 interacts with the *En1* promoter in these tissues (S1A Fig) [33]. These findings raise the possibility that mECE20 may additionally regulate *En1* in these contexts [23,25,26,34].

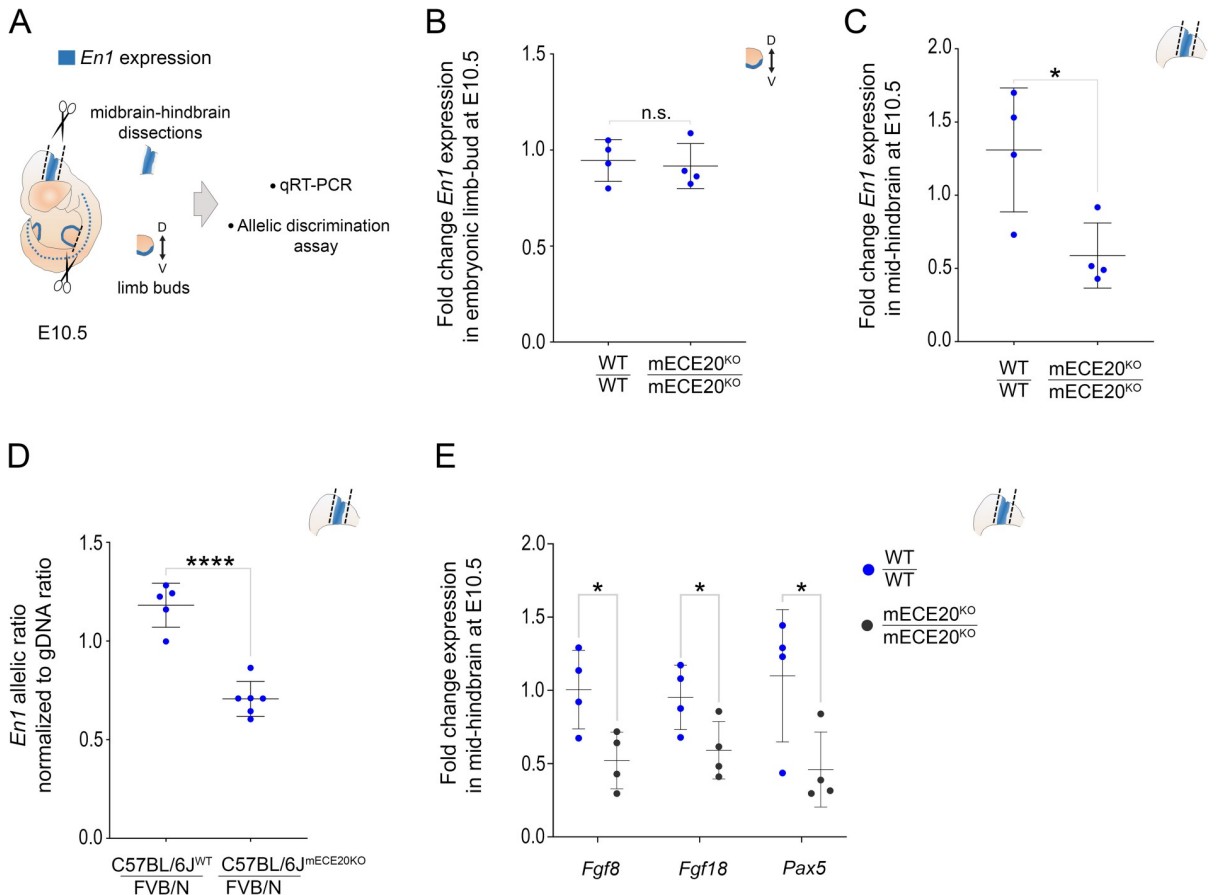

**Fig 3. ECE20 has pleiotropic effects on *En1* expression. (A)** Schematic of mouse embryonic day (E) 10.5 tissues harvested for quantitative analyses of *En1* expression. **(B)** Fold change *En1* mRNA assessed by qRT-PCR from autopods of wildtype (WT / WT), and mECE20$^{KO}$ homozygous (mECE20$^{KO}$/ mECE20$^{KO}$) mice at E10.5 relative to wildtype. **(C)** Fold change in *En1* mRNA by qRT-PCR in midbrain-hindbrain tissue of wildtype (WT / WT), and mECE20$^{KO}$ homozygous (mECE20$^{KO}$/ mECE20$^{KO}$) mice compared to wildtype at E10.5. **(D)** Fold change in *Fgf8*, *Fgf18* and *Pax5* mRNA expression by qRT-PCR in midbrain-hindbrain tissue of wildtype (WT / WT), and mECE20$^{KO}$ homozygous (mECE20$^{KO}$/ mECE20$^{KO}$) mice compared to wildtype at E10.5. **(E)** Normalized *En1* mRNA allelic ratio in midbrain-hindbrain of wildtype (C57BL/6J$^{WT}$ / FVB/N) and mECE20$^{KO}$ (C57BL/6J$^{(mECE20KO)}$ / FVB/N) hybrid mice. Ratios are normalized to the genomic DNA allelic ratio. In **(B-E)** each dot represents the mean value for an individual biological sample and the mean with standard deviation are plotted. In **(B-E)** significance assessed by a two-tailed T-test. ****P<0.0001, * P<0.05, n.s. not significant. (KO) knock-out. In **(B, C)** *Rlp13a* was used as a housekeeping transcript for normalization. See also S3 Fig.

qRT-PCR analysis of normalized *En1* mRNA expression in the ectoderm of embryonic day (E) 10.5 mouse limb buds reveals no significant difference between homozygous mECE20$^{KO}$ and WT sibling controls (Fig 3A and 3B). In line with this, we find no evidence of dorsalization in the limbs of mECE20$^{KO}$ mice. We also find no effect on the expression of ectodermal *En1* in the E10.5 limb buds of mECE18$^{KO}$ mice (S3A Fig). Our findings reveal that despite the evidence of contact, mECE20 is not required for the activation of the *En1* promoter in the limb ectoderm prior to the onset of eccrine gland formation in the region.

In contrast to our findings in the embryonic limb, using qRT-PCR on dissected E10.5 mouse midbrain-hindbrain tissue, we find that normalized *En1* mRNA levels are on average 50% lower in the mECE20$^{KO}$ homozygotes compared to WT sibling controls (Fig 3C). Allele-specific analyses of *En1* transcripts in the E10.5 midbrain-hindbrains of C57BL/6J$^{WT}$: FVB/N and C57BL/6J$^{mECE20KO}$:FVB/N F1 hybrids show a mECE20$^{KO}$-dependent skew in the *En1* allelic ratio demonstrating that mECE20 is a *cis* acting regulatory element required to potentiate

*En1* transcription in this region (Fig 3D). The domain of *En1* expression is unchanged in mECE20^KOs relative to WT controls in this context, indicating that ECE20 regulates the levels rather than the pattern of *En1* expression (S3B Fig). An *in silico* scan for predicted binding sites in ECE20 identifies conserved motifs for a number of brain-enriched transcription factors that may modulate the activity of the enhancer in this context (S3C Fig).

The magnitude of *En1* reduction in the midbrain-hindbrains of ECE20KOs is sufficient to significantly decrease the transcription of *Fgf8*, *Fgf18*, and *Pax 5*, which are in a positive feedback circuit with *En1* and critical for establishing and maintaining the regional identity of this brain region (Fig 3E) [35–39]. In contrast to the significance of disrupting ECE20 on *En1* and the developmental network that regionalizes the embryonic brain, we do not observe any change in the allelic expression of *En1* in the midbrain-hindbrains of E10.5 hybrid mECE18^KO animals (S3D Fig). Our data demonstrate that ECE20 has pleiotropic effects on *En1* expression that are temporally and spatially constrained. Moreover, the contrasting functionalities of the murine homologs of ECE20 and ECE18 in this species give support to a model of differential usage of the two *En1* enhancers in regulating the developmental expression of this gene during mammalian evolution.

### ECE20 is not redundant with ECE18 in the regulation of mouse eccrine gland number

Studies suggest that multiple enhancers often act redundantly to provide robustness to developmental phenotypes [40–42]. Our analyses of the ECE18 enhancer revealed that while this enhancer is required for endogenous *EN1* expression in human basal keratinocytes, its activity is not required for ectodermal *En1* activation in mice [24]. In light of the finding that mECE20 is part of the intrinsic *En1* regulatory machinery in mice, we interrogated whether the absence of discernable eccrine phenotypes in the mECE18^KO may be explained by functional redundancy with the mECE20 enhancer in the eccrine forming skin of this species. Accordingly, we used CRISPR-Cas9 genome editing of mECE20^KO zygotes to generate a compound mECE18^KO; mECE20^KO mutant mouse model (Figs 4A and S4A).

mECE18^KO; mECE20^KO compound homozygous mutant mice are born at Mendelian ratios, are viable and fertile. Comparative analyses of whole-mount epidermal preparations of volar skin reveal no additional reduction in IFP eccrine gland number in mice carrying compound deletions for mECE18 and mECE20 (mECE18^KO; mECE20^KO) as compared to mECE20^KO alone on either a wildtype or in an *En1*-sensitized genetic background (Fig 4B–4D). These data demonstrate that mECE18 is not redundant with mECE20 in the generation of the mouse eccrine gland phenotype and reveal that the *En1* regulatory machinery has diverged to incorporate both shared (ECE20) and lineage- specific (ECE18) modules that control the specification of eccrine gland density in the skin of mammals.

### Discussion

Cis-regulatory modulation of the spatial, temporal, and quantitative expression of the *Engrailed 1* locus is crucial to the formation of eccrine sweat glands in mammals [22,24]. Our findings in the mECE20^KO mouse model, coupled with the pattern of mECE20 activity in transgenic mice, demonstrate that this enhancer is an essential factor controlling *En1* expression in all of these dimensions. While we cannot directly determine the endogenous functional properties of the hECE20 ortholog in human development, the concordance in activity between hECE20 and mECE20 suggests that the role of ECE20 in regulating *En1* is broadly conserved among divergent mammals. This is further supported by the high degree of sequence conservation between mouse and human ECE20 homologs, and contrasts with the

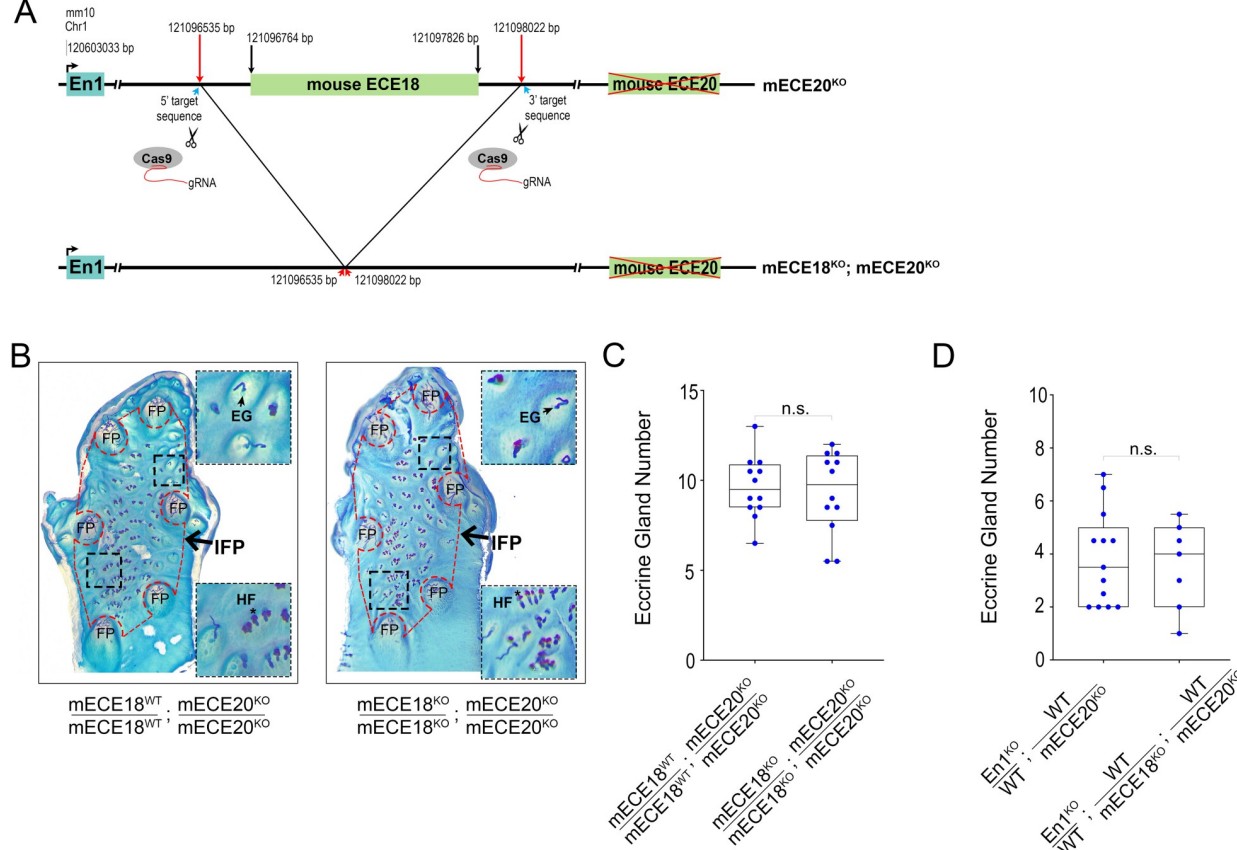

**Fig 4. ECE20 is not redundant with ECE18 in the regulation of mouse eccrine gland number. (A)** Strategy to generate mECE18; mECE20 (mECE18^KO; mECE20^KO) compound knock-out (KO) mouse. **(B)** Representative images of whole mount preparations of the volar hindlimb skin from mECE18^WT/ mECE18^WT; mECE20^KO/ mECE20^KO and mECE18^KO/ mECE18^KO; mECE20^KO/ mECE20^KO adult mice. Epidermal preparations were stained to facilitate quantification of appendages in the interfootpad area (IFP, outlined in red dashed border). Insets show magnified views of regions within the IFP (black dashed border). Footpad (FP), eccrine gland (EG) and hair follicles (HF). **(C)** Quantification of IFP eccrine glands in mECE18^WT/ mECE18^WT; mECE20^KO/ mECE20^KO and mECE18^KO/ mECE18^KO; mECE20^KO/ mECE20^KO mice. **(D)** Quantification of IFP eccrine glands in En1 ^KO/ WT; WT / mECE20^KO and En1 ^KO/ WT; WT / mECE18^KO; WT / ECE20^KO mice. In **(C, D)** each point represents the average number of eccrine glands in the IFP across both hindlimbs of a mouse. In **(C, D)** significance assessed by a two-tailed T-test. n.s. not significant. (KO) knock-out. The median (line) and maximum and minimum are reported for each genotype. See also S4 Fig.

accelerated divergence of the primate specific ECE18 enhancer [24,28]. In light of this evidence for ECE20 functional conservation and because *En1* is the earliest-known factor that is exclusively required for the formation of eccrine glands, ECE20 ranks within the highest tier of the ancestral developmental cascade for making these appendages [17,22]. Future studies to identify the transcription factors that activate ECE20 will thus set the stage for elucidating the mediators that initiate the eccrine developmental program in the ectoderm.

An intriguing observation from our study is that the deletion of ECE20 results in a substantial but not a complete loss of *En1* expression in the basal ectoderm. Consistent with our previous finding that eccrine gland number is dictated by *En1* dosage, eccrine glands are significantly reduced but not entirely absent in mECE20^KO animals. The buttressing of *En1* expression against the loss of mECE20 may reflect the presence of additional *En1* enhancers. Our results showing that both ECE18 and ECE20 human homologs are likely to regulate ectodermal *EN1* during development are certainly consistent with such modular regulation of the locus in primates. Candidate regulatory elements that may provide these additional layers of

mammalian *En1* regulation include the ECE8 (Chr1: 120756823–120757766 [mm10]) and ECE23 (Chr1: 121394405–121395702 [mm10]) elements we have previously identified [24].

The results of our study not only implicate a multi-component network of enhancers that controls ectodermal *En1* in mammalian eccrine gland phenotypes, but also highlight the functional compartmentalization of this network with respect to *En1* regulation in distinct developmental contexts. A clear indication of this is the finding that mECE20 is required to potentiate *En1* expression both in the embryonic brain and also in the skin during eccrine gland development, but not in the limb ectoderm during dorso-ventral patterning. The existence of a separable regulatory module for *En1* expression in the latter context is supported by the identification of the Maenli lncRNA-containing region, deletion of which specifically results in a dorsalization phenotype in both human patients and in mice [27]. Studies of spatio-temporal differences in local chromatin configuration, particularly because of the proximity of *En1* to the TAD boundary, and of the relative availability of transcription factors that regulate the respective enhancers are needed to determine why *En1* engages with different enhancers in different contexts [43]. Considering the developmental importance of maintaining a ventral identity in the limb, it is intriguing to speculate that the evolution of independent modules for ectodermal *En1* regulation in dorso-ventral patterning and in eccrine gland density specification could have facilitated the evolution of generalized eccrine glands in catarrhines.

Beyond its implications for the complexity of the *En1* enhancer network, that ECE20 has pleiotropic effects on *En1* expression not only in the skin but also in the brain, provides a potential explanation for the repeated targeting of the ECE18 enhancer during human evolution. Unlike ECE20, ECE18 activity appears to be restricted to the skin ectoderm. As such, our findings suggest that the potential for incurring deleterious exaptations in addition to the effects on eccrine gland density would be comparably lower from mutations in ECE18 than in ECE20. Future studies are needed to fully characterize the function of ECE20 in the brain, however the observed impact of ECE20 disruption on the EN1/FGF8 neural circuit required for maintaining the regional identities of the midbrain-hindbrain is consistent with such a model. By virtue of its context-specific function, the evolution of the primate ECE18 enhancer would allow catarrhines to generate eccrine gland phenotypic variation at a reduced risk of affecting the development and patterning of the brain. Thus, the pleiotropy of ECE20 coupled with the specificity of ECE18 could not only have favored but also constrained the evolution of catarrhine eccrine gland phenotypes to the ECE18 enhancer. Our findings not only implicate an underlying driver of modularity in ectodermal *EN1* regulation, but also suggest a basis for the extreme derivation of hECE18 and its importance to the evolution of the singular eccrine phenotype of humans.

## Materials and methods

### Ethics statement

All research was approved by the Institutional Animal Care and Use Committee (IACUC) at the University of Pennsylvania Perelman School of Medicine under protocols 806105 and 805904.

### Mice

CD1 (Crl:CD1) timed pregnant and FVB/NCrl mice were purchased from Charles River Laboratories. C57BL/6J mice were purchased from the Jackson Laboratories. En1$^{KO}$ mice were generated in the lab of Dr. Alexandra Joyner [26] and were obtained from the laboratory of Susan Dymecki (Harvard Medical School). En1$^{KO}$ mice were bred onto C57BL/6NTac (Taconic Biosciences) for at least 10 generations. Mice were housed in a vivarium on a 12h light/dark cycle. See also Table 1.

**Table 1. Reagents and resources used in this study.**

| REAGENT or RESOURCE | SOURCE | IDENTIFIER |
|---|---|---|
| Antibodies | | |
| Chicken Polyclonal anti-GFP | Abcam | AB13970 |
| Rabbit Polyclonal anti-Keratin 14 (Poly19053) | Biolegend | REF # 905303 |
| Biotin-SP AffiniPure Rabbit Anti-Chicken IgY (IgG), Fc fragment specific | Jackson ImmunoResearch | REF # 303-065-008 |
| Alexa Fluor 594 | Jackson ImmunoResearch | REF # 711-585-152 |
| Anti-Digoxigenin-AP Fab fragments | Roche | REF # 11093274910 |
| Bacterial and virus strains | | |
| Plasmid: lentiviral stagia GFP-reporter | [24] | N/A |
| Plasmid: mCherry (modified FUtdTW) | Laboratory of Connie Cepko | N/A |
| Plasmid: lentiviral bidirectional vector for luciferase assay | [24] | N/A |
| Chemically Competent *E. coli* | Laboratory of Hao Wu | Strain Stbl3 |
| Biological samples | | |
| Primary human keratinocytes | Core B of the Penn Skin Biology and Diseases and Resource-based Center | N/A |
| Chemicals, peptides, and recombinant proteins | | |
| Polyethylenimine, Linear, MW 25000, Transfection Grade | Polysciences, Inc | REF # 23966–2 |
| Nile Blue A | Sigma-Aldrich | N5632-25G |
| Oil Red O | Sigma-Aldrich | O0625-100G |
| DAPI (4'6-diamidino-2-phenylindole) | Sigma-Aldrich | REF # 10236276001 |
| Keratinocyte serum-free medium (KSFM) | ThermoFisher Scientific | REF # 17005042 |
| Epidermal growth factor (EGF), Human recombinant | ThermoFisher Scientific | REF # 10450–013 |
| Bovine Pituitary Extract (BPE) | ThermoFisher Scientific | REF # 13028–014 |
| Fetal bovine serum (FBS), Hyclone | Fisher Scientific | REF # SH3008003 |
| Trypsin-EDTA (0.25%) | ThermoFisher Scientific | REF # 25200056 |
| Penicillin-Streptomycin (10,000 U/mL) | ThermoFisher Scientific | REF # 15140122 |
| DMEM—Dulbecco's Modified Eagle Medium | ThermoFisher Scientific | REF # 11995065 |
| Critical commercial assays | | |
| VECTASTAIN ELITE ABC Kit, Peroxidase | Vector laboratories | PK-6100 |
| DAB Substrate Kit, Peroxidase (HRP) | Vector laboratories | SK-4100 |
| Dual-Luciferase Reporter Assay System | Promega | E1910 |
| RNeasy Mini Kit | Qiagen | REF # 74104 |
| SuperScript III First-Strand Synthesis System | ThermoFisher Scientific | REF # 18080051 |
| Ribonuclease H | ThermoFisher Scientific | REF # 18021014 |
| RNase-Free DNase set | Qiagen | REF # 79254 |
| PowerUP SYBR Green master mix | ThermoFisher Scientific | REF # A25742 |
| NEBNext High-Fidelity 2X PCR Master Mix | New England Biolabs | REF # M0541S |
| DIG RNA Labeling Kit (SP6/T7) | Roche | REF # 11175025910 |
| Experimental models: Cell lines | | |
| Human: GMA24F1A cell line | Laboratory of Howard Green | [51] |
| Human: HEK293T cell line | Laboratory of Hao Wu | N/A |
| Experimental models: Organisms/strains | | |
| Mouse: CD1/NCrl | Charles River Laboratories | N/A |
| Mouse: C57BL/6J | The Jackson Laboratory | Strain # 000664 |
| Mouse: nu/nu nude mouse | Charles River Laboratories | N/A |

*(Continued)*

**Table 1.** (Continued)

| REAGENT or RESOURCE | SOURCE | IDENTIFIER |
|---|---|---|
| Mouse: En1[KO] | Laboratory of Susan Dymecki | [26] |
| Mouse: mECE20[KO] | This paper | N/A |
| Mouse: mECE18[KO]; mECE20[KO] | This paper | N/A |
| Oligonucleotides | | |
| Primers for cloning, see S1 Table | This paper | N/A |
| Primers for qRT-PCR, see S1 Table | [24] | N/A |
| Primers for allelic discrimination assay, see S1 Table | [22] | N/A |
| Recombinant DNA | | |
| Plasmid: psPAX2 | Laboratory of Didier Trono | Addgene plasmid # 12260 |
| Plasmid: pCL-VSV | Laboratory of Connie Cepko | N/A |
| Software and algorithms | | |
| GraphPad Prism version 9.41 for Windows | Dotmatics | https://www.graphpad.com/ |
| QSVanalyzer software | [52] | http://dna-leeds.co.uk/qsv/ |
| Guide Design Resources | Feng Zhang lab | http://crispr.mit.edu/ |
| Other | | |
| Vevo-2100 Ultrasound | FUJIFILM VisualSonics Inc. | N/A |
| Leica DM5500B microscope | Leica | N/A |
| Leica DEC500 camera | Leica | N/A |
| The Applied Biosystems QuantStudio 7 Flex real-time PCR | ThermoFisher Scientific | REF # 4485701 |
| Spectra Max i3x | Molecular Devices | https://www.moleculardevices.com/products/microplate-readers/multi-mode-readers/spectramax-i3x-readers |

### Generation of lentiviral-mediated transient transgenic mice

The mECE20 lentivirus mediated transgenic mice were generated as previously described [24,44]. Briefly, high-titer of lentiviral particles (mECE20 mixed with mCherry which allows the visualization of infected mice at harvest) were delivered into the amniotic cavity of embryonic day (E) 9 CD1/NCrl mouse embryos (Charles River Labs). Injections were performed under ultra-sound guidance using the Vevo 2100 ultrasound imaging system (Visualsonics, Toronto Canada) equipped with a 35–50 MHz mechanical transducer as described previously [24,44]. Primers used for cloning are reported in S1 Table. All survival surgeries were carried out in accordance with approved IACUC protocols. See also Table 1.

P 2.5 pups were sacrificed and both forelimbs, and both hindlimbs were harvested from mCherry positive animals. Limbs were processed by cryosection and immunohistochemical analysis was performed as described to detect GFP positive clones. See also Table 1.

### Cell culture, transfection, and transduction

HEK293T cells were obtained from the laboratory of Dr. Hao Wu (University of Pennsylvania). Transfection of HEK293T cells was carried out using polyethylenimine (Polysciences Inc.). We used a second-generation packaging system to generate the lentiviruses used in this study (gift from Connie Cepko Harvard Medical School). All viruses were produced in HEK293T cells according to established protocols. Packaging plasmid psPAX2 was a gift from Didier Trono (Addgene plasmid # 12260), envelope plasmid pCL-VSV was a gift from Connie Cepko (Harvard Medical School). See also Table 1.

## Generation of hybrid human-mouse reconstituted skin

The enhancer activity of the hECE20 was tested in a humanized hybrid skin patch assay consisting of human derived epidermis coupled with mouse inductive dermis as described previously [31,45]. This system was previously shown to recapitulate the early stages of human skin development *in vivo* [45]. In brief, inductive mouse dermis was isolated from P0 CD1/NCrl mouse pups as previously described [45]. Primary human keratinocytes were isolated from human neonatal foreskin obtained from Core B of the Penn Skin Biology and Diseases and Resource-based Center. In brief, foreskin was incubated at 4˚C for 12 hours in dispase II (2.4 U/ml). The epidermal sheet was separated from the underlying dermis and then transferred to a 60-mm tissue culture plate. Next, cells were incubated in 0.25% trypsin for 10 min at 37˚C, and then neutralized with 1 ml of fetal bovine serum (FBS). Cell dissociation was carried out using mechanical force against sterile dish. The suspension was passed through a 40-μm strainer and then centrifuged at 200g for 5 min. The cell pellet was resuspended and plated in keratinocyte serum-free medium (SFM) supplemented with human recombinant epidermal growth factor (EGF) and bovine pituitary extract and 1% penicillin-streptomycin (10,000 U/ml) at 37˚C. See also Table 1.

To generate hybrid cysts containing epidermis, dermis, and skin appendages, male nude (nu/nu) mice (Charles River) were anesthetized by isoflurane and injected with a mixture of transfected primary human keratinocytes and mouse dermal cells. A total of 2 x $10^6$ cells (ratio 1:1) were resuspended in a final volume of 100 uL in DMEM (Invitrogen) and injected into the hypodermis of the mouse skin. The injection site was marked by a blue tattoo puncture. Hybrid skin cysts were harvested from mouse backskin 12 days later and fixed overnight in 4% Paraformaldehyde (PFA). Human derived cysts are easily distinguished by their subcutaneous location and nodular appearance. Only human keratinocytes are infected with GFP-expressing, replication incompetent lentivirus allowing for unambiguous distinction between human keratinocytes contributing to the developing epidermis and hair follicles in the cyst and any contaminating mouse keratinocyte cells, which do not have GFP. See also Table 1. Primers to clone hECE20 into the lentiviral vector are listed in S1 Table.

Primary human keratinocytes and HEK293T cells were maintained at 37˚C under 5% $CO_2$. GMA24F1A cells were maintained at 37˚C under 10% $CO_2$.

## Genome alignments

Visualization of genome alignments and genomic annotations was generated by the UCSC Genome Browser (https://genome.ucsc.edu) [46].

## *in silico* motif discovery analysis

DNA binding sites within ECE20 were identified using funMotifs (http://bioinf.icm.uu.se:3838/funmotifs/) with default parameters and the skin and brain datasets [47]. Evolutionary conserved motifs were filtered manually and motif calls were confirmed using the JASPAR 2022 database https://jaspar.genereg.net/ [48].

## *In situ* hybridization, immunohistochemistry, and imaging

Tissues were embedded in OCT (Tissue Tek) and cryo-sectioned at a thickness of 10–12 μm. HRP (Horseradish peroxidase)/DAB(3,3-diaminobenzidine) immunostaining was performed as previously described [24]. Tissue was blocked in PBST (PBS + 0.1% Tween) + 10% normal donkey serum before incubating in chicken anti-GFP primary antibody (1:2000, Abcam). After washing, samples were incubated with biotin-SP-conjugated rabbit anti-chicken

secondary antibody (1:250, Jackson ImmunoResearch). Samples were washed and incubated in Vectastain ABC reagent (Vector laboratories). Enzymatic detection was carried out using the DAB peroxidase substrate kit (Vector Laboratories) according to manufacturer's instructions. Cytokeratin 14 primary antibody (CK14, 1:10000), Alexa Fluor[594] (1:250, Jackson ImmunoResearch) and 4′6-diamidino-2-phenylindole (1:5000, Sigma Aldrich). Whole mount *in situ* hybridization of E10.5 mouse embryos was performed as previously described [49,50]. Briefly, embryos were fixed overnight at 4˚C in 4% PFA. Next day, embryos were dehydrated through a methanol/PBST series. Single-stranded *En1* anti-sense RNA probe was labeled with digoxigenin-UTP (Roche). After hybridization, the labeled RNA probe was detected with anti-digoxigenin-AP Fab fragments (Roche). See also Table 1. Images were acquired on a Leica DM5500B microscope equipped with a Leica DEC500 camera and on a Leica MZ12 stereomicroscope equipped with a Leica DFC295 camara for embryos pictures. See also Table 1.

## Bidirectional luciferase vector and luciferase assays

Quantitative enhancer activity was assessed as previously described in human GMA24F1A human keratinocytes which express the basal keratinocyte marker KRT14 and also express EN1 [24,51]. Briefly, the mouse and human ECE20 orthologs were cloned upstream of a minimal tata-box promoter and upstream of the Firefly Luciferase reporter gene into the bidirectional luciferase lentiviral vector and using the primers listed in S1 Table. Lentivirus production and GMA24F1A human keratinocyte transduction was carried out as previously described [24]. Cells were harvested 72 hours post-transduction, and Firefly and Renilla (for normalization) Luciferase activities were measured using the Dual-luciferase reporter assay system (Promega). Luminescence was detected and quantified on the SpectraMax i3x (Molecular Devices). Experiments were carried out independently at least 3 times in biological triplicate each time. See also Table 1.

## Generation and maintenance of ECE20 knock-out (mECE20[KO]) and of ECE18 and ECE20 compound knock-out (mECE18[KO]; mECE20[KO]) mice

For specific details of targeting design, strategy and validation, screening of founder animals and establishment of mutant lines see S2A (mECE20[KO]) and S4A Figs (mECE18[KO]; mECE20[KO]). In brief, pairs of guide RNAs (gRNAs) were designed using the online tool http://crispr.mit.edu/. Guides were tested to generate deletion of mouse ECE18 or ECE20 *in vitro* in NIH3T3 cells. The reported gRNAs which were confirmed to target mouse ECE18 or ECE20 *in vitro* were used to generate genome edited mice. gRNA selection, generation and *in vitro* testing were performed by the Perelman School of Medicine (PSOM) CRISPR Cas9 Mouse Targeting Core. Validated gRNAs along with Cas9 RNA were microinjected into the cytoplasm of C57BL/6J one cell embryos by the PSOM Mouse Transgenic and Chimeric mouse facility. Mouse ECE18 and ECE20 knock-out junctions were confirmed by Sanger sequencing. All procedures were performed in accordance with approved IACUC protocols. See also Table 1.

## RNA extraction and quantitative RT-PCR

Total RNA extraction, cDNA synthesis and qRT-PCR were performed as previously described [24]. *En1* total expression was determined as previously described and using the following primers S1 Table. Assays were performed in biological replicates consisting of at least three to four pooled, ventral fore or hindlimb skins at P2.5 or embryonic limb-buds at E10.5 from each genotype. Embryonic midbrain-hindbrain dissections were carried out by microdissection. cDNA was generated using SuperScript III (Thermo Fisher) according to the manufacturer's

instructions. qRT-PCR on each biological replicate was carried out using Power SYBR PCR master mix (Thermo Fisher) and each biological sample was analyzed in technical triplicate. Each data point reported in the manuscript represents the mean value for a biological replicate based on the technical triplicate for that sample. qRT-PCR values were normalized to *Rpl13a*. See also Table 1.

### Allelic discrimination assay

*En1* allelic expression assays were performed as previously described [22,24]. Briefly, ventral fore or hindlimb skin consisting of the region containing the five or six volar footpads and intervening IFP were dissected from P2.5 F1 C57BL/6J: FVB/N hybrid mice and RNA was extracted. For the embryonic tissue, limb-buds or mid-hindbrain were dissected at E10.5. Amplification of cDNA and gDNA products was done using the primers listed in S1 Table. *En1* allelic expression was determined by the relative expression of C57BL/6J (mECE20^KO) (mECE18^KO; mECE20^KO) vs. FVB/N allele as distinguished at rs3676156 [22,24]. Allelic expression data was analyzed using the sequencing based QSVanalyzer software [52]. Sequencing was carried out using the *En1* Forward primer listed in S1 Table in technical triplicate for each sample. cDNA was obtained from biological replicates each consisting of three to four pooled, ventral fore or hindlimb skins, or three to four pooled, embryonic limb-buds, or the dissected midbrain-hindbrain regions of an individual mouse, for each genotype. Each data point reported in the manuscript represents the mean value for an individual biological sample. See also Table 1.

### Quantification of eccrine glands

Quantification of mouse interfootpad (IFP) eccrine gland number was performed in whole-mount preparations of the volar skin epidermis of three to four week old euthanized mice as previously described [53]. Whole mount epidermal preparations were stained with Nile Blue (Sigma-Aldrich) and Oil Red O (Sigma-Aldrich) to visualize volar skin appendages (hair follicles and eccrine glands) and hair follicle-associated sebaceous glands, respectively. Every data-point reported in the manuscript represents the average number of IFP eccrine glands in the left and right volar skin of an individual mouse. See also Table 1.

### Statistical analysis

Statistical analysis was performed by means of student's unpaired T-test (two-tailed) using GraphPad Prism version 9.41 for Windows, GraphPad Software, La Jolla California USA, www.graphpad.com. See also Table 1.

### Supporting information

**S1 Fig. Features of the ECE20-containing genomic interval and activity of the ECE20 enhancer in cultured skin cells. (A)** Relative genomic position of Engrailed 1 candidate enhancer (ECE) 20 (boxed) and features of the ECE20-containing topologically associated domain (TAD, solid gray rectangle). Genomic positions of additional positive ECEs (red vertical lines) previously reported in Aldea et al. 2021 [24]. PhastCons scores based on alignment of placental mammals are depicted in green [46]. Called peaks from mouse embryonic limb and midbrain for interaction between the *En1* promoter (anchor) and the genomic region containing ECE20 based on Capture-C (black, horizontal line), for CTCF enrichment (red and blue triangles show site location and directionality of CTCF sites), and for RAD21 enrichment (blue vertical lines) [33]. **(B)** Sequence alignment of mammalian genomes centered on ECE20 using mouse genome build mm10 as the base genome [46]. PhastCons scores for each position

in the alignment are shown in green. Alignment and PhastCons scores are pulled from the USCS Genome browser (http://genome.ucsc.edu) [46] **(C)** Quantitative activity of mouse and human ECE20 orthologs in human GMA24F1A cultured keratinocytes, an immortalized human skin cell line that endogenously expressed *En1* [24,51,54]. The fold change in normalized luciferase activity relative to Control (empty vector) is plotted. Each dot represents a biological replicate. In **(C)** significance is assessed by one-way *ANOVA* and Tukey-adjusted P-values are reported. ***P<0.001, n.s. not significant.
(TIF)

**S2 Fig. Functional characterization of the ECE20 enhancer in ectodermal *En1* regulation and eccrine phenotypes. (A)** Generation of an ECE20 knock-out mouse (mECE20$^{KO}$) by CRISPR-Cas9 mediated genome editing. CRISPR-Cas9 target sequence and genotyping strategy are shown. Deletion junctions were confirmed by Sanger sequencing of F1 pups. **(B)** Fold change in *En1* mRNA by qRT-PCR in P2.5 volar forelimb skin of wildtype (WT / WT), and mECE20$^{KO}$ homozygote (mECE20$^{KO}$/ mECE20$^{KO}$) mice relative to wildtype. **(C)** Normalized *En1* mRNA allelic ratio in volar forelimb of wildtype at P2.5 of wildtype (C57BL/6J$^{WT}$ / FVB/N) and mECE20$^{KO}$ (C57BL/6J(mECE20$^{KO}$) / FVB/N) hybrid mice. Ratios were normalized to the allelic ratio in F1 genomic DNA. Each point represents the mean value across three technical replicates of biological samples consisting of pooled P2.5 volar skins from both forelimbs of two or three mice. **(D)** Location and identity of *in silico*-predicted DNA binding motifs for transcription factors enriched in skin that are evolutionarily conserved between mouse and human ECE20. Relative expression of the cognate transcription factor (TF expression) and motif score (TF score) are shown. Motifs identified using funMotifs (tissue-specific functional motifs) [47]. **(E)** Quantification of interfootpad (IFP) eccrine gland number in adult volar forelimbs of WT / WT and mECE20$^{KO}$/ mECE20$^{KO}$ mice. **(F)** Quantification of IFP eccrine glands in adult, volar forelimbs of En1 $^{KO}$ / WT; WT / WT and En1 $^{KO}$ / WT; WT / mECE20$^{KO}$ mice. In **(B, C)** dots represent an individual biological replicate. In **(E, F)** each point represents the average number of eccrine glands in the IFP across both forelimbs of a mouse. In **(B, C)** mean (line) with standard deviation are plotted. In **(E, F)** the median (line) and maximum and minimum are reported for each genotype. In **(B, C, E, F)** significance assessed by a two-tailed T-test. ***P<0.001, ** *P<0.01, * *P<0.05. (KO) knock-out. In **(B, C)** *Rlp13a* was used as housekeeping transcript for normalization.
(TIF)

**S3 Fig. Characterization of ECE20 and ECE18 in the mouse embryonic midbrain-hindbrain and limb-bud. (A)** Normalized *En1* mRNA allelic ratios in forelimb autopods of wildtype (C57BL/6J$^{WT}$ / FVB/N) and mECE18$^{KO}$ (C57BL/6J$^{mECE18KO}$ / FVB/N) hybrid mice are plotted. **(B)** Whole-mount *in situ* hybridization for *En1* in wildtype (WT / WT) and mECE20$^{KO}$/ mECE20$^{KO}$ mice at E10.5. **(C)** Location and identity of *in silico*-predicted DNA binding motifs for transcription factors enriched in the brain that are evolutionarily conserved between mouse and human ECE20. Relative expression of the cognate transcription factor (TF expression) and motif score (TF score) are shown. Motifs identified using funMotifs (tissue-specific functional motifs) [47]. **(C)** Normalized *En1* mRNA allelic ratios in midbrain-hindbrain of wildtype (C57BL/6J$^{WT}$ / FVB/N) and mECE18$^{KO}$ (C57BL/6J$^{mECE18KO}$ / FVB/N) hybrid mice. In **(A, D)** ratios are normalized to the allelic ratio in genomic DNA, and each point represents the mean value across three technical replicates of a pool of three or four mice for embryonic limb-bud in **(A)**, or individual dissections of midbrain-hindbrain in **(D)** at E10.5. In **(A, D)** the mean (line) and the standard deviation are reported. In **(A, D)** significance assessed by a two-tailed T-test. n.s. not significant. (KO) knock-out.
(TIF)

**S4 Fig. Generation of ECE18; ECE20 compound knock-out mice. (A)** Generation of an ECE18; ECE20 compound knock-out mouse (mECE18$^{KO}$; mECE20$^{KO}$) by CRISPR-Cas9 mediated genome editing. CRISPR-Cas9 target sequence and genotyping strategy are shown. Deletion junctions were confirmed by Sanger sequencing of F1 pups. (KO) knock-out. (TIF)

**S1 Table. Primers used to subclone ECE20 orthologs in mouse transgenic assays.** (DOCX)

**S1 Data. Raw data main figures.** (XLSX)

**S2 Data. Raw data S1–S4 Figs.** (XLSX)

## Acknowledgments

We thank Eric Joyce, Iain Mathieson, Pantelis Rompolas, Paola Kuri, Sixia Huang, and Gabriella Rice for helpful discussions on this project and on the manuscript. We thank Constance Cepko and Clifford J. Tabin for training, technical assistance, and support on ultrasound-guided injection.

Any opinions, findings, and conclusions or recommendations expressed in this material are those of the authors and do not necessarily reflect the views of the NSF. The content is solely the responsibility of the authors and does not necessarily represent the official views of the NIH.

## Author Contributions

**Conceptualization:** Yana G. Kamberov.

**Formal analysis:** Daniel Aldea.

**Funding acquisition:** Heather L. Dingwall, Yana G. Kamberov.

**Investigation:** Daniel Aldea, Blerina Kokalari, Yuji Atsuta, Heather L. Dingwall, Ying Zheng, Arben Nace.

**Methodology:** Daniel Aldea, Yuji Atsuta.

**Project administration:** Yana G. Kamberov.

**Resources:** George Cotsarelis, Yana G. Kamberov.

**Supervision:** Yana G. Kamberov.

**Validation:** Daniel Aldea.

**Visualization:** Daniel Aldea.

**Writing – original draft:** Daniel Aldea, Yana G. Kamberov.

**Writing – review & editing:** Daniel Aldea, Yana G. Kamberov.

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
