## [Decision Letter · Decision Letter 0]

9 Nov 2022

Dear Dr Kamberov,

Thank you very much for submitting your Research Article entitled 'Differential modularity of the mammalian Engrailed 1 enhancer network directs sweat gland development' to PLOS Genetics.

The manuscript was fully evaluated at the editorial level and by independent peer reviewers. The reviewers appreciated the attention to an important problem, but raised some concerns about the current manuscript. Based on the reviews, we will not be able to accept this version of the manuscript, but we would be willing to review a revised version. We cannot, of course, promise publication at that time.

If you decide to revise the manuscript for further consideration at PLOS Genetics, please aim to resubmit within the next 60 days, unless it will take extra time to address the concerns of the reviewers, in which case we would appreciate an expected resubmission date by email to plosgenetics@plos.org.

We are sorry that we cannot be more positive about your manuscript at this stage. Please do not hesitate to contact us if you have any concerns or questions.

Yours sincerely,

David R. Beier

Academic Editor

PLOS Genetics

Bret Payseur

Section Editor

PLOS Genetics

Reviewer's Responses to Questions

**Comments to the Authors:**

Reviewer #1: In this well written paper, Yana Kamberov and co-workers investigate the functions of ECE20, a putative enhancer of the En1 gene. They show that mouse ECE20 drives transgenic reporter gene expression in vivo in volar skin that will develop eccrine glands, as well as in basal epidermis and hair follicles in this region. ECE20 is well conserved between mouse and humans. To interrogate the human ECE20 homolog, the authors generated a lentivirus expressing a reporter gene downstream of human ECE20 and used this to infect primary human keratinocytes. The infected keratinocytes were mixed with inductive dermal cells from mouse backskin and injected subcutaneously into nude mice. The resulting cysts expressed the reporter gene in human basal epidermal and hair follicle cells, which reflects the expression of EN1 In developing human skin. These data indicated that ECE20 is active in both mouse and human skin. To assay for its functional importance, the authors deleted ECE20 in mice using gene editing. This caused reduced levels of En1 expression in volar skin and reduced numbers of sweat glands. Interestingly, En1 transcript levels were also reduced in E10.5 midbrain-hindbrain tissue, but not in E10.5 limb buds, where En1 is also expressed and directs dorso-ventral patterning. These data showed that ECE20 directs both tissue-restricted and temporally restricted En1 expression. The authors previously identified another En1 enhancer, ECE18, that is diverged from the mouse ECE18 sequence. ECE18 is required for En1 expression in human but not mouse skin. To test whether ECE20 and ECE18 function redundantly in mice, the authors generated mice lacking both enhancers. This did not result in a further reduction in En1 expression or eccrine sweat gland number beyond that seen with ECE20 deletion alone.

This paper addresses a very interesting question, namely how gene expression is controlled in a species, tissue and temporal specific fashion by distinct enhancer elements. The experiments were carried out in a logical and rigorous fashion with appropriate controls and statistical analyses. The data are clear and convincing, and will be of interest to a broad range of PLoS Genetics readers, including evolutionary biologists, and researchers interested in enhancer biology and in skin development.

I have several comments on the text; these should be addressed prior to publication to increase the impact of this study and broaden its potential audience:

1. The authors find that ECE20 deletion affects En1 levels in volar skin and in the midbrain-hindbrain region. In volar skin, this causes reduced numbers of eccrine sweat glands. Do the authors also observe abnormalities in the brain? These are predicted from their argument in the Discussion that ECE20 mutation would be detrimental to brain function and therefore selected against.

2. Do the authors observe predicted transcription motifs in ECE20 that are conserved between mouse and human, as would be expected given the apparently similar activity of ECE20 in the two species? Do these motifs differ from those seen in ECE18?

3. Along these lines, are there any transcription factors/transcription factor families predicted to bind conserved motifs in ECE20 that are expressed in the embryonic midbrain as well as in volar skin?

4. The legend for supplemental Figure S1 requires editing for the English.

Reviewer #2: Investigations into gene regulation guiding different tissue morphogenesis is an issue of great interest and complexity. This paper attempts to characterize a new enhancer involved in regulating the expression of engrailed 1 (EN1) that subsequently leads to different downstream consequences.

Engrailed is involved in specifying eccrine sweat glands in mammals and is also involved in patterning the vertebrate mid- and hindbrain of vertebrates and the dorsal-ventral embryonic limb bud. Authors explored candidate Engrailed enhancers. While in humans, ECE 18 has been shown to serve as an enhancer for Engrailed, it does not appear to serve this function in mice. Rather, ECE20, another enhancer identified on of their previous paper, seems to play this role in the mouse. Both ECE18 and ECE20 are found within the human mouse Engrailed topologically associated domain using High-C analysis. mECE20 can induce EGFP expression from a reporter construction appropriately in the mouse ventral autopod, reflecting normal Engrailed expression. A virally transduced hECE20 - eGFP reporter was introduced to human epithelium in a heterotypic skin reconstitution assay with mouse dermis when transplanted to the backs of nude mice. hECE20 can also induce expression eGFP expression in human skin basal keratinocytes in this assay. The ECE20KO studies convincing show the production of phenotypic changes in eccrine gland number. The combined En1KO/WT:WT/mECE20KO experiments further confirm the role of ECE20KO in regulating eccrine gland number. They then use the ECE18WT/mECE20KO experiments demonstrate that mECE18KO has no additional effects on endocrine gland number in mouse limbs. So, in the mouse, ECE20 plays this unique role. It is interesting the authors note that the inhibition by ECE20KO is not complete and other enhancers may also be involved in this regulation.

Enhancer study is an important question but is also complex. Here authors have developed new mice and use new functional approaches to assess the role of Engrailed enhancers. They have gain valuable new knowledges. The following can be clarified and discussed without new experiments.

1. Was High-C performed independently on the mouse dorsal limb and brain domains at P2.5 and P10.5? It is possible that these two regions temporal as well as regional differences in TAD configurations that might account for the differential response. Please clarify.

2. The viral enhancer-reporter constructs test human and mouse ECE20 function without regard to the TAD configuration. While it shows that human and mouse ECE20 are capable of regulating gene expression in the limb and skin, it doesn’t necessarily mean that they perform this function in their native state. Please discuss.

v

Minor

• ECE should be defined in abstract.

**Have all data underlying the figures and results presented in the manuscript been provided?**

Reviewer #1: Yes

Reviewer #2: Yes

PLOS authors have the option to publish the peer review history of their article (what does this mean?). If published, this will include your full peer review and any attached files.

Reviewer #1: No

Reviewer #2: No

---

## [Decision Letter · Decision Letter 1]

13 Jan 2023

Dear Dr Kamberov,

We are pleased to inform you that your manuscript entitled "Differential modularity of the mammalian Engrailed 1 enhancer network directs sweat gland development" has been editorially accepted for publication in PLOS Genetics. Congratulations!

Yours sincerely,

David R. Beier

Academic Editor

PLOS Genetics

Bret Payseur

Section Editor

PLOS Genetics

Comments from the reviewers (if applicable):

Reviewer's Responses to Questions

**Comments to the Authors:**

Reviewer #1: The authors have responded satisfactorily to my comments and this paper is now acceptable for publication in PLoS Genetics.

Reviewer #2: The revision is satisfactory for this reviewer.

**Have all data underlying the figures and results presented in the manuscript been provided?**

Reviewer #1: Yes

Reviewer #2: Yes

PLOS authors have the option to publish the peer review history of their article (what does this mean?). If published, this will include your full peer review and any attached files.

Reviewer #1: No

Reviewer #2: No

**Data Deposition**

http://datadryad.org/submit?journalID=pgenetics&manu=PGENETICS-D-22-01084R1

**Press Queries**

---

## [Editor Report · Acceptance letter]

1 Feb 2023

PGENETICS-D-22-01084R1 

Differential modularity of the mammalian Engrailed 1 enhancer network directs sweat gland development 

Dear Dr Kamberov, 

We are pleased to inform you that your manuscript entitled "Differential modularity of the mammalian Engrailed 1 enhancer network directs sweat gland development" has been formally accepted for publication in PLOS Genetics! Your manuscript is now with our production department and you will be notified of the publication date in due course.

With kind regards,

Anita Estes

PLOS Genetics

On behalf of:
